# FBXW7 Gene Mutation and Expression in Colorectal Cancer (CRC): A Systematic Review from Molecular Mechanisms to Clinical Translation

**DOI:** 10.3390/ijms262311318

**Published:** 2025-11-23

**Authors:** Giulia Arrivi, Gabriella Gentile, Michela Roberto, Donatella Delle Cave

**Affiliations:** 1Oncology Unit, Department of Clinical and Molecular Medicine, Sant’Andrea University Hospital, Sapienza University of Rome, 00189 Rome, Italy; 2Department of Experimental Medicine, Sapienza University of Rome, Viale Regina Elena, 324, 00161 Rome, Italy; g.gentile@uniroma1.it; 3Department of Radiological, Oncological and Pathological Science, Sapienza University of Rome, 00185 Rome, Italy; 4Oncology Unit, Policlinico Umberto I, Haematological, Oncological and Dermatological Department, Sapienza University of Rome, 00185 Rome, Italy; mikiroberto87@gmail.com; 5Institute of Genetics and Biophysics ‘Adriano Buzzati-Traverso’ (IGB), National Research Council (CNR), 80131 Naples, Italy

**Keywords:** FBXW7, FBW7, colon cancer, colorectal cancer, next-generation sequencing (NGS), molecular profiling

## Abstract

In the context of precision oncology, understanding the molecular drivers of colorectal cancer (CRC) is critical for improving prognosis and guiding targeted therapy. FBXW7 is a tumor suppressor that plays a pivotal role in CRC by regulating the degradation of key oncogenic proteins, influencing tumor initiation, growth, therapeutic response, and metastatic behavior. Mutations in FBXW7 occur in 6–10% of CRC. Despite its biological relevance, the prognostic and predictive role of FBXW7 in CRC remains unclear, with inconsistent findings across studies. This systematic review collects and analyzes current evidence on FBXW7 mutations and expression in CRC, emphasizing its potential role in risk stratification, therapeutic response, and personalized treatment approaches. A total of 113 records were selected on PubMed, SCOPUS, Web of Science and Cochrane Central Register of Controlled Trials from 2015 and January 2025, of which 48 examined the preclinical landscape of FBXW7 in CRC and 65 focused on its clinical role. FBXW7 mutations are associated with different clinicopathological patterns, including early-onset disease, microsatellite instability, and co-occurring driver alterations, all of which shape prognosis and treatment outcomes. While some variants correlate with immune infiltration and better survival, others, especially when co-mutated, predict aggressive disease and poor outcomes. Furthermore, FBXW7 alterations contribute to chemoresistance and anti-EGFR therapy resistance but also reveal potential therapeutic vulnerabilities. These findings underscore FBXW7’s promise as a prognostic biomarker and a potential target for precision oncology strategies in colorectal cancer.

## 1. Introduction

In the era of precision medicine, understanding tumor biology continues to evolve, providing increasingly refined insights into the molecular, biochemical, and cellular characteristics of cancer. Advances in next-generation sequencing (NGS) and the declining cost of genomic analysis have accelerated comprehensive molecular profiling across malignancies, including colorectal cancer (CRC) [1].

Although the identification of genomic alterations has deepened our understanding of carcinogenesis, the functional consequences of post-transcriptional and post-translational regulation remain incompletely understood. From genome to proteome, multiple layers of control contribute to biological complexity and phenotypic diversity [2]. Dysregulation of protein degradation pathways—leading to oncoprotein stabilization—has emerged as a critical mechanism in tumorigenesis [3].

F-box and WD repeat domain containing 7 (FBXW7) is one of the key components of the ubiquitin ligase called Skp1-Cullin1-F-Box (SCF) complex and is involved in the degradation of many oncoproteins through the ubiquitin-proteasome system (UPS). Acting as a tumor suppressor, FBXW7 is expressed in three isoforms (α, β, γ), encoded by a single locus on chromosome 4q32 and is pivotal in maintaining cellular homeostasis and genomic stability [4].

Notably, FBXW7 controls the expression of multiple oncoproteins, including Cyclin E1, transcriptional factors c-Myc and c-Jun, Notch receptor, SREBP (a coactivators of PARPγ) and transcriptional factor PGC-1α, thereby suppressing tumor development [5,6]. Inactivation of FBXW7 through genetic mutation, genomic deletion, or promoter hypermethylation represents a major driver of carcinogenesis [7,8]. The gene displays a distinctive spectrum of mono-allelic missense mutations, typically affecting conserved arginine residues within the β-propeller domain crucial for substrate binding [9].

According to COSMIC data, FBXW7 mutations occur in approximately 7.8% of all human cancers, with the highest prevalence in cholangiocarcinoma (35%), T-cell acute lymphoblastic leukemia (31%), endometrial (9%), and gastric cancers (6%). In CRC, mutation rates range from 6 to 10%, placing FBXW7 among the most commonly altered tumor suppressor genes [8]. Targeted DNA-based NGS remains the most reliable method for detecting these mutations, while immunohistochemistry (IHC) and quantitative real-time polymerase chain reaction (RT-PCR) are valuable for assessing expression levels. The most frequent variants include S582L (19.3%), R465H (16.6%), R505C (14.9%), and R479Q (14.9%), with roughly half of all alterations being missense or nonsense substitutions. Approximately 50% of FBXW7 mutations in large intestine cancers are missense or nonsense mutations, with similar distributions between frameshift insertions and deletions [10,11].

Differences in FBXW7 status have been noted between primary and metastatic CRC. Alterations were identified in 13% of primary tumors versus 7% of peritoneal metastases, with enrichment in metachronous liver metastases from left-sided CRCs (11%) [12,13].

Functionally, FBXW7 loss promotes drug resistance by stabilizing anti-apoptotic proteins such as Mcl-1, reducing sensitivity to 5-fluorouracil (5-FU), oxaliplatin, regorafenib, and sorafenib [13,14].

FBXW7 polymorphisms—particularly the rs6842544 variant—are associated with an increased risk of CRC, with mutations leading to decreased FBXW7 expression in tumor tissues [15]. Collectively, these data underscore the multifaceted role of FBXW7 in CRC pathogenesis, prognosis, and therapy response, highlighting its potential as a biomarker for risk assessment and therapeutic targeting.

While the prognostic impact of mismatch-repair deficiency (MMR) and RAF/RAS alterations is well established in clinical practice, the relevance of other molecular abnormalities, including FBXW7, remains uncertain. Despite extensive investigation, studies on FBXW7 mutations and expression have yielded inconsistent results. Furthermore, comprehensive analyses dedicated to CRC are limited.

This review aims to systematically organize and evaluate the available evidence to clarify the clinical relevance of FBXW7 in CRC, with particular focus on its prognostic and predictive value and its emerging role in personalized treatment strategies.

## 2. Materials and Methods

### 2.1. Search Strategy

This systematic review was conducted according to the Preferred Reporting Items for Systematic Reviews and Meta-analysis (PRISMA) guidelines [16]. In January 2025, we performed a comprehensive literature search in PubMed, SCOPUS, Web of Science and Cochrane Central Register of Controlled Trials to identify available published articles that investigated the role of FBXW7 gene mutation and/or expression in carcinogenesis or cancer development and its possible prognostic and predictive role in colorectal cancer. For PubMed the searching strategy was as follows: (“fbxw7” OR “fbw7”) AND (“colon cancer” OR “colorectal cancer”); while for SCOPUS, it was: “fbxw7 OR fbw7 AND colon cancer” limited to “Medicine, Biochemistry, Genetics and Molecular Biology”, “Multidisciplinary”, “Pharmacology”, and “Toxicology and Pharmaceutics” areas.

The key words “fbxw7, fbw7, colon” were used for searching in Web of Science limited to ”Oncology”, “Biology”, “Chemistry”, “Chemistry Analytical and Medicinal” “Cell Biology”, “Biochemistry Molecular Biology”, “Pathology”, “Gastroenterology Hepatology”, “Surgery”, “Medicine Research Experimental”, “Physiology” and “Pharmacology”, while “fbxw7, fbw7, colon cancer” for searching in Cochrane Central Register of Controlled Trials.

### 2.2. Study Selection Criteria and Data Extraction

Studies investigating the role of the FBXW7 gene and its alterations in the carcinogenesis, progression, and chemoresistance of CRC, as well as those evaluating the clinical significance of FBXW7 mutations and expression in human CRC, were considered eligible for inclusion in this review. The literature search was restricted to English-language publications and limited to animal and human preclinical studies, as well as retrospective or prospective clinical trials. Meta-analyses, review articles, case reports, case series, editorials, and commentaries were excluded. Three authors (G.A., G.G., and D.D.C.) independently screened the identified records, and full-text articles were reviewed for all studies that met the inclusion criteria. Any disagreements or uncertainties were resolved through discussion. A formal risk-of-bias or study quality assessment was not performed due to the heterogeneity of the included studies, which encompassed descriptive preclinical and clinical investigations with differing aims and endpoints. Following this selection process, relevant data were extracted and entered by the authors into a predesigned Microsoft Excel form.

## 3. Results

### 3.1. Literature Search and Included Studies

A total of 1991 records from four databases—PubMed, Scopus, Web of Science and Cochrane Library—has been identified with the aforementioned search strategy, and 605 were excluded before screening by automation tools. The three authors (G.A., G.G. and D.D.C.) independently screened, by title and abstract, 1386 records: 1047 for irrelevant content and 187 duplicates were excluded. Of the remaining 152 reports assessed for eligibility, we excluded 2 case reports/series, 3 comments/perspectives, 12 reviews or meta-analysis, 1 guideline, 1 non-English article and 20 for irrelevant topic after a more in-depth reading. The selection process is illustrated in Figure 1 and resulted in a thorough analysis of 113 records published between 2015 and January 2025, of which 48 examined the preclinical landscape of FBXW7 in CRC and 65 focused on its clinical role.

### 3.2. Preclinical Studies

#### 3.2.1. FBXW7 in CRC and Tumorigenesis

FBXW7 is the substrate recognition component of the Skp1-Cdc53/Cullin–F-box–protein complex (SCF/β-TrCP) [4,5,6]. The SCF complex mediates the ubiquitination of target proteins as the final step before their degradation in the ubiquitin–proteasome system (UPS). Within this complex, FBXW7 recognizes substrates that have been phosphorylated at specific residues within a conserved CDC4 phosphodegron (CPD) motif. This phosphorylation is carried out by glycogen synthase kinase 3 beta (GSK3β) [4,5,6]. Once phosphorylated, the substrate is recruited by FBXW7 to the SCF complex, where it undergoes polyubiquitination. The polyubiquitinated substrate is then delivered to the 26S proteasome for proteolytic degradation.

The FBXW7 gene plays a complex role in tumorigenesis: both its overexpression and its inactivating mutations have been implicated in cancer development, and numerous preclinical studies have investigated its function in this context (Table 1).

FBXW7 is a key regulator of cell proliferation. In mice, the overexpression of FBXW7 in mice was found in the transit-amplifying progenitor cell compartment, and its deletion resulted in impaired goblet cell differentiation and accumulation of highly proliferating progenitor cells. Therefore, loss of FBXW7 promotes tumor growth and blocks differentiation, effects that are in part driven by aberrant activation of Wnt signaling. Cells lacking both FBXW7 and the mitotic checkpoint exhibit not only reduced proliferation but also a significant increase in aneuploidy, likely due to premature anaphase entry [17]. Accumulation of the DEK proto-oncogene has been observed in FBXW7-mutant tumors. In particular, increased DEK expression enhances cell division and alters tropomyosin (TPM) RNA splicing, potentially influencing cell proliferation [18]. To further characterize the FBXW7-associated genes in CRC, the co-expression genes predicted by cBioPortal online analysis using Spearman’s correlation are reported in Appendix A.

Many preclinical studies investigated the effect of miRNAs binding FBXW7 in CRC, although the specific mechanisms are still unclear. miR-223, upregulated in CRC tissue, might bind to the FBXW7 gene and block its expression, promoting CRC cell proliferation and inhibiting apoptosis via the Notch and Akt/mTOR signaling pathways [19]. miR-27a can directly downregulate the tumor suppressor FBXW7 and promote cell proliferation. miR-27a knockdown and the subsequent increase in FBXW7 protein levels inhibits NOTCH, JUN and MYC signaling, and this causes CRC differentiation [20]. The interplay between FBXW7 and miRNAs was demonstrated by the downregulation of FBXW7 thorough miR-92b, which itself is negatively regulated by c-MYC. This suggests a potential positive regulatory feedback loop in CRC between c-MYC, miR-92b, and FBXW7 [20]. Moreover, by upregulating FBXW7, miR-92b-3p suppression inhibited CRC proliferation, invasion, and migration [21].

FBXW7 significantly impacts metastatic behavior. Its depletion induces epithelial–mesenchymal transition (EMT), thereby promoting invasion and migration [22]. In vitro, ex vivo, and in animal models of metastasis, the FBXW7-Zinc-finger E-box-binding homeobox-2 (ZEB2) axis affects critical characteristics of cancer cells, including stemness/dedifferentiation, chemoresistance, and cell migration [23]. In addition, the FBXW7–Drosophila caudal-related homeobox transcription factor 2 (CDX2) interaction contributes to loss of differentiation and enhanced malignancy. There may be an inverse relationship between CDX2 levels and tumor stage, as loss of CDX2 expression has been shown in a number of CRC cell lines. Through two phosphodegron motifs found within CDX2, FBXW7 promotes CDX2 ubiquitination and degradation in a GSK3β-dependent manner, resulting in decreased CDX2 expression and activity in colon cancer cells. Several miRNAs, as miR-182 and miR-503 contributed to the malignant transition of colon adenoma to adenocarcinoma, cooperating in tumor suppressor FBXW7 control [24].

Several studies highlight FBXW7 as a guardian of genomic stability. Codeletion of both FBXW7 and p53 induces highly penetrant, aggressive, and metastatic adenocarcinomas in gut epithelial cells, and allografts generated from these tumors form extremely malignant adenocarcinomas [25]. BUBR1, a component of the mitotic spindle assembly checkpoint, was identified as potential synthetic lethal candidates for FBXW7. In addition, CENtromere Protein A-mediated centromere activation of cyclin E1/CDK2 in conjunction with FBXW7 deletion to induce chromosomal instability and tumor growth [26]. FBXW7 mutations contribute to carcinogenesis. The most common alterations are mono-allelic missense mutations affecting critical arginine residues required for substrate binding. Both missense and null mutations in FBXW7 induce widespread transcriptional reprogramming by altering transcription factor occupancy and chromatin activation. Moreover, cancer-specific FBXW7β mutations and the resulting loss of FBXW7 function enhance de novo fatty acid synthesis and lipogenesis to meet the heightened biosynthetic demands of tumor growth and progression [10,11,27].

FBXW7 also contributes to antitumor immunity. In microsatellite-stable (MSS) CRC, the nuclear proto-oncogene SET is frequently overexpressed [28]. Gao and colleagues demonstrated that SET promotes immune evasion by two complementary mechanisms: first, SET enhances DNA MMR protein levels, preserving the MSS “cold” phenotype; second, SET blocks FBXW7 from binding to c-Myc, thereby preventing FBXW7-mediated ubiquitination and proteasomal degradation of c-Myc. Stabilized c-Myc in turn represses CCL5 expression, reducing CD8^+^ T-cell recruitment into the tumor microenvironment. These findings identify FBXW7 as a critical mediator of antitumor immunity in MSS CRC, and suggest that inhibiting SET to restore FBXW7 function, in combination with immune checkpoint blockade, offers a promising strategy for otherwise immune checkpoint blockade–refractory MSS CRC [28]. Using a large cohort of colorectal cancer organoids (CCOs) that faithfully preserve primary tumor genetics, including TP53, KRAS, APC and crucially FBXW7 mutations, Cho and colleagues defined a cancer-intrinsic immuno-genomic profile marked by high HLA-II mRNA expression, which strongly predicts superior overall and recurrence-free survival. FBXW7 mutations, like other key genomic alterations, were maintained in CCOs and found to associate with specific tumor immune microenvironment (TIM) classes: all MSI-H organoids fell into the “Exhausted” group (predicting likely response to PD-1/PD-L1 blockade), while organoids with activated Wnt/β-catenin signaling, often driven by FBXW7 loss, clustered in the “Desert” group characterized by immune exclusion. Thus, FBXW7 status not only influences cancer cell–intrinsic immunogenic features but also helps shape the surrounding TIM, underscoring its dual role in tumor progression and immunotherapy responsiveness [29].

Boretto et al. used a CRISPR–Cas9 strategy to establish a human colon-organoid biobank harboring the most frequent FBXW7 hotspot mutations, R278, R479, R479Q, R505C, Y545C, S582L and R689W, with R479 and R505 being the predominant targets in CRC [30]. Unlike conventional carcinoma cell lines, these organoids maintain near-physiological genomic stability over long-term culture. Strikingly, all FBXW7 mutants exhibited a ~10,000-fold reduction in their EGF requirement for sustained growth; C-terminal mutations showed a milder effect, suggesting they retain partial E3-ligase function. Mechanistically, mutant organoids accumulated EGFR protein despite unchanged EGFR mRNA levels, a consequence of diminished ubiquitination of the phosphorylated receptor. These results implicate FBXW7 status as a key determinant of EGFR turnover and signaling, underscoring its potential as a biomarker to guide EGFR-targeted therapies in colorectal cancer [30]. Moreover, FBXW7 deficiency stimulates the production of colon cancer stem-like cells in tumor-sphere culture [31]. All the articles that investigated the role in FBXW7 in CRC tumorigenesis have been collected in Table 1.

**Table 1 ijms-26-11318-t001:** Preclinical studies on the role of FBXW7 in CRC tumorigenesis.

Author, Year	Type of Study	Mechanism of FBXW7 Studied	Biological Effect on Tumorigenesis Mediated by FBXW7
Bialkowska AB 2014 [32]	Genomic analysis of CRC cell lines	Oncogenic role for the P301S KLF5 mutant through interaction with FBXW7α	KLF5 P301S mutation blocks FBXW7-mediated degradation, increasing its stability and activity.
Chan DKH 2023 [33]	WT and mutant FBXW7 CRC cells cocultured using a Transwell system Mass spectrometry	AKAP8 mediates DNA damage transfer from FBXW7-mutant to neighboring wild-type cells, affecting local subclonal populations.	AKAP8 is secreted in the microenvironment of FBXW7 mutant cells
Luan RG 2025 [34]	Clinical observational study analyzing protein expression in 38 colon cancer tissue samples.	FBXW7 mediates ubiquitination and degradation of cyclin E1, correlated with phosphorylation at Ser73 and Thr395.	Promotes degradation of cyclin E1, potentially inhibiting cell cycle progression and tumor growth in colon cancer.
Huber AL 2016 [35]	Experimental study using primary mouse fibroblasts, genetic knockout mice (Cry2−/−), and human cancer cell lines to investigate CRY2 function.	Although FBXW7 is a standard SCF adaptor, the study shows that CRY2–FBXL3 forms its own SCF complex that degrades c-MYC independently of FBXW7.	FBXW7 normally limits MYC-driven proliferation by degrading c-MYC, but CRY2–FBXL3 provides a separate MYC-degradation pathway; losing CRY2 therefore increases MYC activity and promotes tumorigenesis.
Duhamel S 2016 [36]	Comprehensive study using epithelial cells, imaging, gene silencing, drug inhibition, and a transgenic mouse model.	ERK1/2 signaling downregulates Fbxw7β, preventing Aurora A’s ubiquitin-mediated degradation and causing its accumulation.	Loss of Fbxw7β leads to Aurora A buildup, causing cytokinesis failure, polyploidy, and chromosomal instability that promote genomic instability and cancer progression.
Mu Y 2017 [37]	Study combining CRC patient tissues with functional assays in HCT116 and SW480 cell lines.	FBXW7 suppresses Notch1 signaling, and FAM83D knockdown increases FBXW7 levels, thereby reducing Notch1 activity.	Higher FBXW7 levels inhibit CRC progression by reducing proliferation, migration, invasion, and promoting apoptosis.
Grim JE 2012 [25]	Mouse model study examining the effects of jointly deleting FBXW7 and p53 in intestinal epithelium.	FBXW7 acts as a tumor suppressor by degrading multiple oncogenic proteins, helping control proliferation, differentiation, and genomic stability.	FBXW7 loss alone alters differentiation and oncogene levels, but combined with p53 loss drives aggressive, metastatic intestinal cancer.
Li QG 2018 [38]	Study of 276 CRC tissues with in vitro validation using IHC, molecular assays, and functional analyses.	FBXW7 functions as an E3 ubiquitin ligase targeting HIF1α for degradation, thereby reducing CEACAM5 (CEA) expression in a HIF1α-dependent manner.	FBXW7 suppresses CRC progression by inhibiting tumor cell migration and metastasis, leading to improved overall survival and disease-free survival.
Zhan P 2015 [39]	Experimental study of colon cancer using proteomics and molecular assays.	FBXW7 targets ENO1 for GSK3β-dependent ubiquitin-proteasomal degradation.	FBXW7 inhibits colorectal cancer by downregulating ENO1, reducing CCL20, lactate, proliferation, and migration.
Lu HR 2019 [40]	Experimental study of CRC using clinical tissues, cell lines, and mouse xenografts.	Circ-FBXW7 acts as a tumor-suppressive RNA by regulating NEK2, mTOR, and PTEN pathways.	Circ-FBXW7 suppresses colorectal cancer by inhibiting proliferation, migration, invasion, and tumor growth.
Babaei-Jadidi R 2011 [18]	Conditional knockout mouse study using Apc^Min/+^ crosses with histological, molecular, and proteomic analyses.	FBXW7, as part of the SCF^FBXW7^ complex, degrades oncogenic proteins; its loss causes aberrant signaling and oncogene accumulation.	FBXW7 loss promotes intestinal tumorigenesis by enhancing proliferation, disrupting differentiation, causing DEK accumulation, altering TPM splicing, and accelerating adenomas with APC deficiency.
Li, N 2015 [41]	Experimental study of colorectal cancer using cell lines, mouse models, and human tissue validation.	FBXW7 indirectly modulates p53 phosphorylation at Ser15/Ser18 via upstream kinases like CK1α.	FBXW7 loss causes phospho-p53(Ser15) accumulation, alters TP53 gene expression, and drives chemoresistance, promoting colorectal cancer progression.
Wei W 2023 [27]	Experimental study of colorectal cancer using cell lines, organoids, and mouse/xenograft models with molecular, biochemical, and metabolic analyses.	FBXW7β promotes FASN degradation via K48-linked ubiquitination, while CSN6 stabilizes FASN by degrading FBXW7β; GSK3β phosphorylation of FASN enables FBXW7β recognition.	FBXW7β suppresses colorectal cancer by reducing FASN, lipid accumulation, and tumor growth; its loss promotes FASN stability and tumor progression.
Long, YP 2019 [42]	In vitro co-culture study of colon cancer cells and macrophages, validated by RNA-seq and qRT-PCR.	FBXW7α controls macrophage polarization through the miR-205/SMAD1 axis; its loss shifts M1/M2 balance.	FBXW7α suppresses TAM M1 polarization; its knockdown boosts inflammatory cytokines, potentially inhibiting colorectal cancer.
Iwatsuki M 2010 [43]	Clinical observational study combined with in vitro functional assays.	FBXW7 regulates degradation of oncogenic proteins c-MYC and cyclin E via ubiquitin-mediated proteolysis; its suppression leads to accumulation of these proteins.	Loss of FBXW7 promotes tumor progression by enhancing cell proliferation, invasion, and poor prognosis in colorectal cancer, acting as a tumor suppressor.
Liu Z 2021 [19]	Study integrating bioinformatics (TCGA, GTEx), CRC tissues, and in vitro experiments in HCT116 cells.	miR-223 directly inhibits FBXW7 by binding its 3′UTR, reducing mRNA and protein levels.	FBXW7 downregulation enhances CRC cell proliferation and inhibits apoptosis through Notch and Akt/mTOR activation.
Chen Y 2024 [44]	Study integrating GEO dataset analysis, CRC tissues, and in vitro experiments in HCT116 and Caco-2 cells.	miR-25-3p directly targets FBXW7 3′UTR, decreasing its mRNA and protein levels.	FBXW7 loss promotes CRC cell proliferation and survival via oncogenic protein accumulation, while its overexpression suppresses growth and induces apoptosis.
Gong L 2018 (abstract) [21]	In vitro study to analyze the effect on CRC invasion of miR-92b-3p, a target of FBXW7	miR-92b-3p downregulates FBXW7 in CRC, and blocking it or overexpressing FBXW7 limits tumor growth and spread.	Inhibiting miR-92b-3p suppresses CRC proliferation, invasion, and migration by upregulating FBXW7, highlighting its role in carcinogenesis and metastasis.
Ou BC 2016 [45]	Study integrating clinical tissue analysis with in vitro and in vivo functional experiments.	Plk2 binds to FBXW7 and promotes its proteasome-mediated degradation, leading to stabilization and accumulation of Cyclin E.	Degradation of FBXW7 by Plk2 increases Cyclin E levels, promoting cell proliferation, inhibiting apoptosis, and enhancing colorectal tumor growth.
Wang YL 2013 [31]	In vitro study using FBXW7 knockout colon cancer cell lines.	FBXW7 regulates EMT, migration, invasion, and stemness via mTOR signaling.	FBXW7 loss promotes EMT, migration, invasion, and stemness, which are counteracted by mTOR inhibition.
Guo Z 2012 [46]	In vitro study using human CRC cell lines SW620, HT29 and HCT116	Rictor associates with FBXW7 to form an E3 complex participating in the regulation of c-Myc and cyclin E degradation.	Rictor, within the FBXW7 complex, drives c-Myc and cyclin E degradation, and impaired signaling leads to their accumulation in CRC cells
Li L 2014 * [24]	Study of human colon tissues, CRC cell lines, and knockout models using molecular and histological analyses.	Sequential upregulation of miR-182 and miR-503 promotes colon adenoma-to-adenocarcinoma progression by jointly suppressing FBXW7.	Blocking both miR-182 and miR-503 in HCT116 colon cancer cells resulted in increased FBXW7 expression and significantly reduced tumor size in xenograft models
Khan OM 2018 [47]	In vitro and in vivo study identifying FBXW7 interactors via proteomics in CRC cells, validated by Usp9x deletion in mouse intestines.	The deubiquitinase USP9X antagonizes FBXW7 ubiquitylation, thereby stabilizing FBXW7 protein and regulating c-MYC degradation.	Usp9x inactivation reduced secretory cell differentiation, increased progenitor proliferation and tumor burden, while c-Myc heterozygosity mitigated tumor formation in Usp9x-deficient mice
Lee YS * [48]	Analysis of 232 CRC patient tissues for tumor-associated macrophage subtypes, complemented by in vitro, in vivo, and ex vivo studies on PI3Kγ regulation	M1 and M2 macrophages exerted opposite effects on CRC progression, mediated via the FBXW7-MCL-1 axis, while PI3Kγ inhibition in macrophages influenced EMT and cytotoxicity in tumor cells.	Patients with a lower M2/M1 ratio (<3) experienced significantly better progression-free and overall survival; targeting macrophage PI3Kγ suppressed EMT features and enhanced colon cancer cell death, supporting its value as an immunotherapeutic target.
Lin L 2020 [49]	Experimental study in CRC and normal epithelial cells assessing Trametinib and TRAIL effects using siRNA, transfection, viability, colony formation, apoptosis, Western blot, qPCR, and co-immunoprecipitation.	FBXW7 functions as an E3 ligase that promotes proteasomal degradation of Mcl-1 phosphorylated by GSK-3β.	Trametinib enhances TRAIL-mediated apoptosis in CRC cells by promoting FBXW7-dependent Mcl-1 degradation, thereby boosting its anticancer effect.
Bengoechea-Alonso MT 2010 [50]	In vitro study examining TGIF1 phosphorylation and degradation under altered Fbxw7 activity.	FBXW7 targets phosphorylated TGIF1 for ubiquitin-mediated proteasomal degradation.	Loss of FBXW7 function leads to accumulation of phosphorylated TGIF1 and suppression of TGFβ-dependent transcriptional activity.
Kumar Y 2016 [22]	In vitro study of FBXW7-mediated CDX2 ubiquitination using coimmunoprecipitation and phosphodegron mutations in colon cancer and HEK293T cells.	FBXW7, in a GSK3β-dependent manner, binds to two phosphodegron motifs on CDX2 via its WD domain, promoting CDX2 ubiquitination and proteasomal degradation	FBXW7 and GSK3β overexpression reduces CDX2 levels and function, causing growth arrest in colon cancer cells; disruption of both CDX2 phosphodegrons prevents FBXW7-mediated degradation.
Bajpai S 2022 [51]	Study combining molecular and cellular experiments with protein interaction, ubiquitination assays, and conditional knockout mice.	FBXW7 functions as an E3 ubiquitin ligase that recognizes phosphorylated c-Myc at T58/S62, mediating its polyubiquitination and proteasomal degradation.	FBXW7 limits c-Myc accumulation, thereby suppressing uncontrolled cell proliferation and acting as a tumor suppressor.
Diefenbacher ME 2015 [52]	In vivo and in vitro study using MEFs, intestinal epithelial cells, and double knockout mouse models.	FBXW7 targets oncogenic substrates for degradation, while USP28 opposes this by deubiquitinating them, including in FBXW7-deficient cells.	USP28 loss in FBXW7-deficient mice slows proliferation, restores differentiation, and partially reverses tumorigenesis, showing USP28 promotes tumor growth without FBXW7.
Eun JC et al., 2021 [29]	Translational ex vivo study of 87 patient-derived CRC organoids combining multi-omic profiling and drug sensitivity testing.	FBXW7 loss in CRC associates with CD8^+^ T-cell exhaustion, UPR activation, WNT/β-catenin dependency, and heightened sensitivity to PPAR inhibition, revealing potential therapeutic vulnerabilities.	FBXW7 loss promotes tumorigenesis by driving immune evasion (CD8^+^ T-cell exhaustion) and proteostatic stress (UPR), while creating WNT/CTNNB1-dependent vulnerabilities (e.g., FH535 sensitivity).
Boretto et al., 2024 [30]	Study using engineered human CRC organoids with CRISPR base editing, proteomic and transcriptomic analyses, and drug response testing.	FBXW7 normally degrades phosphorylated proteins like EGFR; mutations impair this, causing EGFR and other oncogene accumulation.	FBXW7 mutations reduce EGF dependency, enhance MAPK/EGFR signaling, promote proliferation, and confer resistance to anti-EGFR therapies, thereby driving tumor progression.
Gao et al., 2025 [28]	Preclinical experimental study combining bioinformatics analyses, in vitro assays, and in vivo mouse models of CRC.	FBXW7 promotes c-Myc degradation, while SET stabilizes c-Myc by competing for FBXW7 binding.	FBXW7 suppresses MSS CRC tumorigenesis by degrading c-Myc, reducing mismatch repair proteins, immune evasion, and the immunosuppressive microenvironment.

Abbreviations: CRC: colorectal cancer; WT: wild type; MSS: microsatellite stable. Footnotes: * also clinical results.

#### 3.2.2. FBXW7 and Drug Resistance

FBXW7 demonstrated to have anti-drug resistance actions via different substrates and related pathways. The majority of these substrates are oncogenic proteins frequently overexpressed in CRC. The FBXW7 role on drug resistance is mediated by its ubiquitination and degradation of transcription factors, cell cycle regulators and antiapoptotic proteins, and by activation of Notch and m-TOR pathways. Loss of FBXW7 sensitizes cancer cells to some drugs, while developing resistance to other types of chemotherapies [14] (Table 2). Chemotherapeutic drugs, such as Oxaliplatin and 5-fluorouracil (5-FU), are currently and frequently used in CRC management. Cryptochrome 2 (CRY2) is a circadian clock protein involved in the cell cycle, regulated by FBXW7. FBXW7 suppresses CRY2 via ubiquitination and increases the susceptibility of CRC cells to chemotherapy. In CRC patient samples, low FBXW7 expression is associated with high CRY2 level and major sensitivity to oxaliplatin. Furthermore, low FBXW7 expression is associated with poor survival [53]. Some circular RNAs have been found to be associated with CRC chemoresistance. Circular RNA F-box and WD repeat domain containing 7 (circ-FBXW7) was detected as weakly expressed in oxaliplatin-resistant CRC cells, but microRNA (miR)-18b-5p was upregulated in oxaliplatin-resistant CRC cells. Circ-FBXW7 has been discovered as a target of miR-18b-5p. Interestingly, circ-FBXW7 administration via exosomes could reduce oxaliplatin resistance in CRC by directly binding to miR-128-3p, implying a viable therapeutic option for oxaliplatin-resistant CRC patients [54]. A further intriguing method for examining and determining the cellular response mediated by FBXW7 to different chemotherapies is the intestinal organoid system. Organoids lacking in FBXW7 exhibited increased resistance to 5-FU. Higher tolerance to 5-FU is a result of FBXW7-mutated cancer cells’ higher rate of proliferation, resistance to the terminal differentiation and cell death [55]. A member of the cytokinin family, N6-isopentenyladenosine (IPA) has been shown to exhibit anticancer activity against a variety of human epithelial cancer cells, including CRC. In vitro models with varying FBXW7 mutational statuses, IPA influenced the growth of CRC by upregulating the expression of FBXW7. IPA and 5-FU work in powerful synergy in FBXW7 wild type cells to suppress the growth of tumor cells [56]. Regorafenib, a multi-kinase inhibitor targeting RAS/RAF/MEK/ERK signaling, has been shown to increase overall patient survival and was approved for metastatic CRC patients after the other standard treatments [57]. Myeloid cell leukemia 1 (Mcl-1), one member of the Bcl-2 family that promotes survival, is often overexpressed in a variety of malignancies. FBXW7 determines regorafenib sensitivity through proteasomal degradation of Mcl-1. Deleting FBXW7 in FBXW7-wild-type CRC cells abolishes Mcl-1 degradation and restores drug resistance phenotypes of FBXW7-mutant cells [58]. Paclitaxel is one of the most commonly used chemotherapeutic agents used for multiple cancer treatments. Overexpression of FBXW7 sensitized CRC cells to Paclitaxel with a negative correlation between FBXW7 and glucose metabolism. NADPH oxidase 1 (Nox1), a superoxide-generating NADPH oxidase, is negatively regulated by FBXW7. Indeed, Nox1 promotes the Paclitaxel resistance and glucose metabolism of CRC [59]. DYRK2 (Dualspecificity tyrosine-phosphorylation-regulated kinase 2) is a Ser/Thr kinase that affects FBXW7 stability in response to DNA damage and sensitivity to chemotherapeutic drug such as Doxorubicin or Paclitaxel [60].The microRNA (miR)-223/FBXW7 pathway has also been implicated in the mechanism of doxorubicin chemoresistance in CRC. Overexpression of miR-223 lowered FBXW7 level and doxorubicin sensitivity in CRC cells [61]. Furthermore, FBXW7 mutational status seems to play a role as a biomarker for Heat shock protein 90 (Hsp90)-targeted therapy in CRC. Inactivating mutations of FBXW7 are intrinsically insensitive to Hsp90 inhibitors [62]. Radioresistance is a significant contributor to poor treatment outcomes in CRC. The underlying mechanism of radioresistance, however, is unclear [63]. MicroRNA-19b (miR-19b) and FBXW7 seem to be involved in the regulation of CRC radioresistance. FBXW7 has been identified as a possible target of miR-19b. MiR-19b was increased in radiation-resistant cells via downregulating FBXW7 expression. The restoration of FBXW7, on the other hand, abolished the effects of miR-19b on radioresistance [64].

### 3.3. Clinical Studies in CRC

#### 3.3.1. Correlation Between FBXW7 Alterations and Demographic/Clinic-Pathological Features of CRC

The distribution of FBXW7 mutations across various demographic and clinic-pathological features of CRC remains controversial. Although mutations in FBXW7 have been linked to tumor aggressiveness and poor prognosis, particularly in adolescent and young adult (AYA) patients, they are also frequently detected in older individuals, likely reflecting the accumulation of endogenous deamination events over time [74].

Tricoli et al. reported that FBXW7 mutations were more frequent in AYA patients than in adults, suggesting a distinct molecular profile that may underlie the more aggressive behavior of early-onset CRC [75]. Similarly, Kothari et al. found a ~2.5-fold higher frequency of FBXW7 mutations in younger versus older patients (27.5% vs. 9.7%, *p* = 0.0022), which may influence screening and treatment strategies for this age group [76].

Early-onset CRC more often exhibits microsatellite instability-high (MSI-H), high tumor mutational burden (TMB) and concurrent mutation in ARID1A (60%), ATM (60%), FAT1 (60%), FBXW7 (60%), and KMT2B (60%), whereas MSS tumors typically harbor alterations in TP53 (75%), APC (68%), and KRAS (48%). This pattern highlights MSI-H early-onset CRC as a distinct subset enriched for FBXW7 alterations [77].

A microbiota-based classification identified three oncomicrobial CRC subtypes (OCS1–3) in stage I to IV CRCs. The subtype 1 (OCS1)—defined by Fusobacterium nucleatum and oral pathogens—was associated with MSI-H features, CIMP positivity, right-sided location, high grade, and BRAF V600E/FBXW7 mutations. These molecular and microbial features correlated with poor prognosis in OCS1 MSS tumors [78].

Racial and ethnic disparities have also been described in early-onset CRC. In non-hypermutated colorectal cancer (<17.78 mutations/Mb), FBXW7 mutation rates were 17% in young Asian or Pacific Islander patients, 11% in non-Hispanic Whites, and 10% in non-Hispanic Blacks [79]. Conversely, Escobar et al. observed a higher incidence of FBXW7 mutations in older CRC patients compared with those harboring CTNNB1 mutations (median age 61.1 vs. 52.4 years, *p* = 0.037) (median age of 52.4 vs. 61.1, *p* = 0.037) [74]. FBXW7 mutations were mutually exclusive with CTNNB1 (*p* < 0.0001) and more common in left-sided/MSS cancers, often co-occurring with TP53 and KRAS alterations, consistent with previous findings in stage II CRC [80].

No significant difference in sidedness and mutational frequency of FBXW7 was found in stage I to IV Chinese and Indian CRC cohorts; however, in the latter, 50% of patients who developed metastases harbored FBXW7 mutations. In terms of the correlation between gene mutation and gender, the frequency of mutations affecting FBXW7 was higher in male patients (*p*  =  0.01063) [23,72,73].

Somatic FBXW7 frameshift mutations have been reported in 9% of hereditary nonpolyposis colorectal cancers (HNPCC) and single-base substitutions in 9% of familial adenomatous polyposis (FAP) carcinomas. FBXW7 alterations were also detected in adenomas (6% in FAP, 4% in sporadic cases), supporting its role in early tumorigenesis in both hereditary and sporadic CRC [81].

#### 3.3.2. FBXW7 Alterations in Localized CRC

Several evidence support the involvement of FBXW7 mutations—alongside TP53 and alterations in RTK–RAS signaling pathways—in the progression from normal colonic epithelium to adenoma and carcinoma, suggesting its potential as an early marker of colorectal carcinogenesis and tumor progression through subclonal dysregulation of tumor suppression and DNA repair pathways [82,83]. In CRC, FBXW7 mutations are frequently observed, though their clinical implications vary across populations and disease stages. Mutations appear enriched in early-stage, organ-confined CRC compared with metastatic disease; patients without distant metastases show a higher incidence of FBXW7 and NOTCH pathway alterations [84].

Focusing on the difference by ethnicity of the populations studied, a study on 534 Japanese stage III CRC samples confirmed an overall spectrum of mutations similar to that of the TCGA in Western population but revealed a higher FBXW7 mutation rate (18.5%) than in Western populations [75,76]. In Chinese cohorts, frequencies ranged from 9.9% in stage II–III CRC to 15.9% across stages I–IV following primary resection [85,86]. Among Thai patients with stage II–III CRC, the frequency was 14.8% (predominantly missense), while Taiwanese populations showed 18.75% across stages I–III [87]. In Brazilian and African American cohorts, mutation rates were 10.9% and 4%, respectively, with mutual exclusivity reported between FBXW7 and NOTCH2 mutations [80,88].

Although several research suggested a potential association between the mutational status of FBXW7 and prognosis in Western and mixed populations, this topic remains highly debated and controversial [89]. Recent data failed to demonstrate prognostic significance in localized CRC treated with adjuvant therapy [90] and no independent prognostic value was observed for FBXW7 mutations in stage II/III CRC in the VICTOR trial [91]. Similarly, a study evaluating the effect of XELOX adjuvant therapy on MACC1, FLNA, and FBXW7 found no significant change in gene expression before or after treatment (*p* > 0.05), excluding these as prognostic biomarkers in that setting [92].

Conversely, in a retrospective analysis of relapsed stage II CRC patients, FBXW7 mutations occurred only in KRAS wild-type tumors, and co-mutations of TP53 and FBXW7 were associated with significantly shorter disease-free survival (DFS) and overall survival (OS) [80]. This was confirmed in a retrospective cohort including localized and advanced CRC, where FBXW7 mutations correlated with reduced DFS in stage III–IV disease (*p* = 0.002), although mutually exclusive TP53 and FBXW7 alterations were also noted [93].

While stage II FBXW7 wild-type patients showed a trend toward better survival, no significant OS difference was observed overall between mutant (7.5%) and wild-type (92.5%) tumors in stage I–III CRC. However, hotspot mutations (R465H/R465C/R479Q) were associated with MSI-H status (*p* = 0.025), earlier stage (*p* = 0.045), and improved OS [10]. In this cohort FBXW7 mutated tumors were significantly more likely to be MSI-H compared to wild-type (16.7% vs. 10.0%, *p* = 0.025). Another comprehensive study confirmed the link between FBXW7 mutations, higher TMB, increased MSI, and immune cell infiltration, suggesting a potential immunoregulatory role [89].

This association was further validated in peripheral blood samples from localized MSI CRC, where recurrent mutations in MLL3, NOTCH1, FBXW7, PIK3CA, and ERBB4 were detected pre- and post-surgery [94]. In contrast a Chinese cohort showed higher FBXW7 mutation rates in MSS versus MSI tumors, although sample-size imbalance limits interpretation [95].

Clinicopathological correlations reveal that FBXW7 gene fusions often co-occur with dMMR in stage II–III CRC, particularly in poorly differentiated (*p* = 0.019), hepatic-flexure tumors (*p* < 0.001) [96]. Furthermore, FBXW7 was identified as significantly upregulated in non-perineural invasion (NPNI) compared to perineural invasion in stage II CRC, suggesting a potential role as a biomarker for PNI [97]. This concept has been reinforced by the evidence that FBXW7 expression was significantly reduced in CRC tissues compared to adjacent normal tissue (*p* < 0.001), with low expression correlating with increased lymph node metastasis (*p* < 0.001), advanced TNM stage (*p* < 0.001), and worse survival in Chinese cohort [98].

Non-invasive detection of FBXW7 mutations is emerging as a clinically relevant tool. Circulating tumor DNA (ctDNA) analysis using next-generation sequencing (NGS) can identify FBXW7 alterations associated with recurrence after surgery, suggesting its use for postoperative risk stratification [99]. Similarly, stool DNA assays have detected FBXW7 mutations in ~17% of stage I–III CRC cases, with a mutational profile concordant with matched tumor tissue, reinforcing stool-based molecular testing as a non-invasive diagnostic and prognostic approach [100].

Overall, current data highlight the complex, context-dependent role of FBXW7 in CRC progression. While findings suggest its promise as a prognostic marker and potential therapeutic target in localized CRC, its clinical validity remains unconfirmed.

#### 3.3.3. FBXW7 Alterations in mCRC

The acquisition of mutational profile of FBXW7, as previously mentioned, has emerged as a key factor in the tumorigenesis, prognosis and spread of metastatic colorectal cancer (mCRC).

Alterations in FBXW7 and its downstream NOTCH signaling pathway have been reported more frequently in patients without organ metastases compared with those with extra-regional spread [101,102,103], suggesting that the FBXW7/NOTCH axis may play a role in metastatic development and could serve as a predictive marker for organ-specific dissemination [103].

In matched analyses of primary and metastatic CRC samples, de novo FBXW7 mutations were detected in metastatic lesions (particularly in lung and liver) but not in corresponding primaries, indicating that these alterations may represent late genetic events linked to metastatic adaptation [104]. Their emergence underscores the importance of sequencing metastatic sites to capture the complete mutational landscape and guide treatment planning. In this contest, selecting the most appropriate biopsy site is therefore critical for prognostic stratification and treatment strategy.

Most mCRC studies report a high genetic concordance for APC, KRAS, FBXW7, PIK3CA, BRAF, SMAD4, and ACVR2A, between primary tumors and liver, lung, nodes, and brain metastatic sites [105,106]. However, discrepancies have been observed for ovarian and peritoneal metastases, where FBXW7 alterations were more common in primary CRCs than in peritoneal lesions (13% vs. 7%, *p* < 0.008) [13].

Clinicopathologic correlations reveal a higher FBXW7 mutation rate in rectal primaries (10.5%) compared with left- (5.2%) and right-sided (2.7%) tumors, and an association with lung metastasis biology [107]. Consistent findings indicate that co-occurring TP53 and FBXW7 mutations are enriched in liver metastases from left-sided CRC, particularly in metachronous presentations [12].

FBXW7 mutations have also been reported more frequently in patients abstaining from alcohol (6.8% vs. 2.3%, *p* = 0.035) and in those with a body mass index (BMI) ≤ 25 kg/m^2^ (7.1% vs. 1.5%, *p* =  0.016) [108]. In early-onset mCRC (<50 years), pooled TRIBE/ TRIBE2 analyses showed higher FBXW7 (19.6% vs. 9.8%, *p* = 0.05) and POLE (3.9% vs. 0%, *p* = 0.03) mutation rates, though chemotherapy intensification did not produce age-specific survival differences [109].

Co-mutation analyses have identified correlations between PTEN and FBXW7 (*p* = 0.015) and between FBXW7 and BRAF (*p* = 0.009) in formalin-fixed mCRC specimens from the Kingdom of Saudi Arabia [110]. In mismatch repair–proficient (pMMR) tumors, FBXW7 alterations appear as late evolutionary events, potentially reflecting adaptive responses rather than primary oncogenic drivers. Detection of concurrent FBXW7 and ERBB2 mutations after neoadjuvant therapy suggests that these may arise under therapeutic pressure or represent expansion of resistant clones [111].

Emerging evidence indicates that FBXW7 alterations, particularly when acquired during or after treatment, may contribute to resistance to cytotoxic and targeted agents by fostering the survival and expansion of therapy-resistant clones. Thus, FBXW7 is considered a promising biomarker for predicting chemotherapy response, though not yet clinically validated. Acquisition of FBXW7 mutations has been described as an early mechanism of resistance to anti-EGFR monoclonal antibodies, often co-occurring with KRAS, NRAS, BRAF, and PIK3CA mutations. These combined alterations predict treatment non-responsiveness and shorter progression free survival (PFS), emphasizing the need for early detection and molecular monitoring [97,98,99,100]. Longitudinal ctDNA analyses during systemic therapy demonstrate that rising FBXW7 mutant allele fractions correlate with shorter PFS and impending progression, supporting comprehensive ctDNA-based monitoring also in mCRC management [112,113].

Significant intratumoral heterogeneity in FBXW7, PIK3CA, and KRAS expression has been documented between primary tumors and synchronously resected liver metastases in KRAS wild type mCRC during targeted treatment [114], suggesting that FBXW7 may promote both metastatic establishment and tumor heterogeneity, potentially impacting therapeutic response [115].

In the specific context of colorectal liver metastasis (CRLM), FBXW7 mutations (5.7%) were strongly associated with poorer OS (*p* = 0.015), with 5-year OS of 40.4% in mutant vs. 59.4% in wild-type tumors [116]. Low FBXW7 expression after hepatectomy was related to lower DFS (3-year DFS 12.5% vs. 47.0%, *p* = 0.023) and identified as an independent predictor of recurrence (hazard ratio = 2.39, *p* = 0.017) [101].

Larger cohort analyses have confirmed missense FBXW7 mutations as negative prognostic factors, with median OS of 28.7 months for mutants vs. 46.6 months for wild type [117]. In CAIRO/CAIRO2 clinical trials, poor-prognosis subtype 3 CRCs were enriched for BRAF (*p* < 0.0001) and FBXW7 (*p* = 0.01) mutations [118,119,120,121]. In a first-line phase II biomarker study (WJOG7612GTR) baseline ctDNA-detectable FBXW7 mutations correlated with inferior outcomes on oxaliplatin/fluoropyrimidine plus bevacizumab, suggesting treatment resistance [122].

Despite its relatively low mutation frequency (~6%), FBXW7 remains biologically relevant for molecular classification, though it has not yet defined prognostically distinct mCRC subsets [123]. Similarly, a 14% mutation rate in Chinese cohorts did not translate into independent prognostic value, underscoring the context-dependent and co-mutation–driven impact of FBXW7 [124]. Notably, co-occurring TP53 or SMAD4 mutations amplify the adverse prognostic effect of FBXW7, suggesting synergistic mechanisms in tumor aggressiveness [125].

## 4. Discussion

FBXW7 occupies a central position in tumor biology. As the substrate-recognition subunit of the SCF E3 ubiquitin ligase complex, it regulates the proteasomal degradation of numerous proteins controlling proliferation, differentiation, and cell fate. FBXW7 loss-of-function produces complex, context-dependent effects: it can promote tumor aggressiveness by stabilizing oncogenic substrates (e.g., Cyclin E, EGFR) yet also generate pharmacologically exploitable vulnerabilities. This duality—driving tumor progression while creating “druggable dependencies” illustrates both the challenge and potential of FBXW7 as a therapeutic and prognostic marker.

Preclinical data consistently show that FBXW7 deficiency causes accumulation of oncogenic proteins without corresponding mRNA increases, reflecting impaired ubiquitin-mediated degradation. CRISPR-engineered colon organoids demonstrate reduced dependence on exogenous EGF and decreased sensitivity to EGFR–MAPK inhibition, suggesting post-transcriptional adaptive resistance. The GSK3β–FBXW7 axis has also been linked to HIF-1α/VEGF-A–mediated angiogenesis, though its clinical significance in CRC remains unclear.

Not all FBXW7 variants exert equivalent effects. Hotspot mutations within the WD40 domain (notably R465 and R479) differentially impair substrate recognition and may explain discordant clinical outcomes. Population-level and molecular heterogeneity further complicate interpretation: mutations appear enriched in adolescents and young adults with MSI-H/TMB-high tumors and co-mutations in ARID1A, ATM, or FAT1 but have also been reported in older patients through age-related mutational processes [74,75,76]. Microbiome-defined subtypes add another layer—an oral-pathogen/Fusobacterium-rich group (OCS1) shows association with BRAF V600E and FBXW7 mutations and poorer outcomes in MSS CRC [78]. Sidedness and ethnicity may also influence prevalence, with higher rates observed in some East Asian and left-sided/MSS cohorts [74,79,126,127].

In localized CRC, the prognostic impact of FBXW7 remains unsettled. Several large datasets—including adjuvant-treated cohorts such as VICTOR—did not identify FBXW7 as an independent prognostic marker (e.g., VICTOR) [10,92]. Conversely, other studies suggest its impact depends on molecular context: concurrent FBXW7 and TP53 mutations correlate with worse disease-free and overall survival, whereas MSI-H tumors with hotspot FBXW7 variants (e.g., R465H/R479Q) may show favorable outcomes and increased immune infiltration [10]. Reduced FBXW7 expression correlates with adverse clinicopathologic features (lymph-node metastasis, advanced TNM stage, perineural invasion) and poorer survival in some Chinese cohorts, and FBXW7 gene fusions have been reported alongside dMMR in poorly differentiated, hepatic-flexure-predominant tumors [78,98].

Non-invasive detection through ctDNA and stool DNA has identified FBXW7 mutations correlating with recurrence risk stage I–III disease, supporting their potential utility for surveillance [100].

Taken together, these data support variant-level reporting (e.g., R465C vs. R465H), comprehensive annotation (MSI/TMB/co-mutations), and integration of stool/ctDNA assays for risk stratification and surveillance—while recognizing that, outside clinical trials, FBXW7 alone should not currently alter adjuvant decisions.

In mCRC, FBXW7 often emerges as a late evolutionary event. De novo mutations may arise in metastases (liver/lung) despite absence in the primary, emphasizing the value of metastasis-directed sequencing and longitudinal ctDNA monitoring to track clonal evolution, overcoming the intratumoral and intersite heterogeneity of FBXW7 mutations [104].

Although overall concordance between primary and metastatic sites is generally high, discordance has been reported for ovarian and peritoneal lesions, reinforcing the need for site-specific sampling and ctDNA integration [105,106,128]. Clinically, FBXW7 mutations and low expression correlate with worse OS/DFS and higher recurrence risk after hepatectomy [116], and several large cohorts have identified missense FBXW7 variants as independently prognostic for shorter OS [117]. However, these associations are inconsistent across populations, and detrimental effects appear strongest when co-alterations occur in TP53 or SMAD4 [125].

About a therapeutic point of view, FBXW7 loss may mediate both primary and acquired resistance. Emerging evidence implicates FBXW7 mutations as an early mechanism of anti-EGFR resistance, often co-occurring with KRAS, NRAS, BRAF, ERBB2 or PIK3CA alterations and linked to reduced PFS [129]. Rising FBXW7-mutant ctDNA alleles during first-line therapy, surgery or radiotherapy correlate with shorter PFS and impending progression, supporting comprehensive, serial ctDNA assessment to guide on-treatment decisions [122]. Preclinical models suggest that MCL-1 inhibition or combined MAPK/PI3K blockade could restore drug sensitivity in FBXW7-mutant CRC. Additionally, FBXW7-dependent ubiquitination of PD-1 has been reported, raising the hypothesis that FBXW7 status might influence immune infiltration and checkpoint inhibitor response, particularly in MSI-H/TMB-high tumors, though this remains to be validated clinically [130,131]. Overall, FBXW7 is a promising—though not yet clinically validated—biomarker in CRC. The relationships between its molecular pathways and clinical outcomes are still incomplete and variably supported across studies. To provide a clearer overview, Table 3 summarizes the main FBXW7-related mechanisms and their corresponding clinical implications in CRC.

Current evidence—derived from heterogeneous, largely retrospective and preclinical studies—remains insufficient for clinical application. Standardized assays, validated IHC cut-offs, and large, prospective, molecularly annotated cohorts are needed to define its true prognostic and predictive value. Future research should focus on hotspot-specific effects, longitudinal tissue/ctDNA integration, and co-mutation-based stratification to clarify how FBXW7 alterations can be incorporated into precision oncology frameworks.

## Figures and Tables

**Figure 1 ijms-26-11318-f001:**
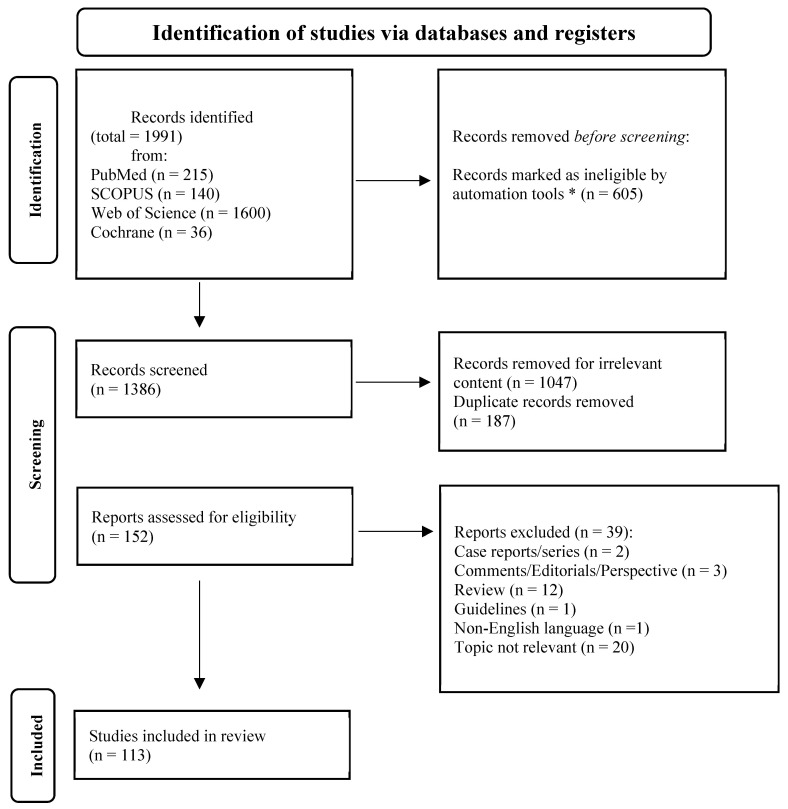
PRISMA 2020 flow diagram for new systematic reviews which included searches of databases and registers only. Legend: * SCOPUS search limited to “Medicine, Biochemistry, Genetics and Molecular Biology”, “Multidisciplinary”, “Pharmacology”, and “Toxicology and Pharmaceutics” areas; WEB OF SCIENCE limited to” Oncology”, “Cell Biology”, “Chemistry Analytical and Medicinal” “Cell Biology”, “Biochemistry Molecular Biology”, “Pathology”, “Gastroenterology Hepatology”, “Surgery”, “Medicine Research Experimental”, “Physiology” and “Pharmacology Pharmacy” and only articles were selected.

**Table 2 ijms-26-11318-t002:** Preclinical studies on the role of FBXW7 in CRC therapy resistance mechanisms.

Author, Year	Type of Study	Mechanism of FBXW7 Studied	Therapy Resistance Mechanisms Mediated by FBXW7
Fang L 2015 [53]	In vitro	CRY2 and FBXW7 expression are inversely correlated, influencing chemoresistance.	FBXW7 promotes CRY2 degradation, increasing CRC cells’ sensitivity to oxaliplatin.
Tong J 2017 [58]	In vitro/in vivo (Mice)	Role of GSK3β/FBXW7-dependent Mcl-1 degradation in killing of CRC cells by Hsp90 inhibitors	GSK3β/FBXW7-driven Mcl-1 degradation is essential for Hsp90 inhibitor–induced apoptosis in CRC; FBXW7 status and Mcl-1 stability determine drug response and guide patient stratification and combination therapies.
Ding J 2018 [61]	In vitro	Role of miR-223/FBXW7 pathway in chemosensitivity in different CRC cell lines	High FBXW7 enhances doxorubicin sensitivity in CRC, and miR-223 reduces FBXW7 levels, decreasing drug sensitivity.
Fiore D 2019 [56]	In vitro In vivo (Mice)	N6-isopentenyladenosine role in CRC proliferation with different mutational status of FBXW7 and TP53 genes	In FBXW7- and TP53-wild type CRC cells, N6-isopentenyladenosine synergizes with 5-Fluorouracil, suggesting FBXW7 restoration as a strategy for personalized therapy.
Xu Y 2021 [54]	In vitro/ In vivo (Mice)	Role of Circular RNA F-box and WD repeat domain containing 7 (circ-FBXW7) in CRC oxaliplatin resistance	Circ-FBXW7 is downregulated in oxaliplatin-resistant CRC, and exosomal circ-FBXW7 can reduce resistance by targeting miR-128-3p.
Wang HP 2021 [59]	In vitro	Biological roles and mechanisms of FBXW7 in taxol resistance	FBXW7 sensitizes CRC cells to Taxol by suppressing Nox1 and glucose metabolism, and its downregulation drives resistance, highlighting the FBXW7–Nox1–metabolism axis as a therapeutic target
Sun T 2022 [64]	In vitro In vivo (Mice)	Roles of exosomal microRNA-19b (miR-19b) in CRC radioresistance.	CRC-derived exosomes promote radioresistance and stemness via miR-19b, which targets and downregulates FBXW7.
Huang G 2023 [65]	In vivo (pts 30) in vitro	FBXW7-Nox1-mTOR pathway and Cisplatin chemoresistance of CRC	FBXW7 was negatively associated with CRC tumor recurrence and cisplatin resistance and positively associated with CRC patient survival rates.
Homma S 2019 [66]	In vivo (55 CRC patients receiving neoadjuvant therapy)	The role of FBXW7 as a cell cycle mediator was investigated, focusing on its expression in colorectal cancer stem cell populations and its impact on cell cycle arrest and chemoresistance	High FBXW7 expression correlated with poor pathological response to treatment, increased chemoresistance in cancer stem cell subsets, and FBXW7 knockdown enhanced sensitivity to anticancer drugs both in vitro and in vivo.
Izumi D 2017 [67]	In vitro	Upregulation of FBXW7 leads to c-Myc degradation, promoting colorectal CSC resistance to chemotherapy-induced cell cycle arrest.	FBXW7 is specifically upregulated in CSCs after drug treatment, decreasing c-Myc and conferring drug resistance; FBXW7 knockdown increases drug sensitivity, with findings confirmed in liver metastases from CRC patients post-chemotherapy.
Ruan F 2024 [68]	Only abstract In vitro/in vivo	Clitocine upregulates FBXW7 by inhibiting A2B receptor/cAMP/ERK signaling, enhancing FBXW7-mediated MCL-1 degradation.	Clitocine decreases intracellular cAMP via competitive binding to A2B receptor, enhancing FBXW7 expression through reduced promoter DNA methylation, leading to increased MCL-1 degradation and improved drug sensitivity; A2B knockout or cAMP analog treatment reverses these effects.
Belmonte-Fernández A 2023 [69]	In vitro	FBXW7 regulates DNA damage response and cisplatin sensitivity by ubiquitinating the MRN complex.	Cisplatin induces FBXW7-dependent MRN degradation through the autophagy-lysosome pathway, promoting tumor cell death, while lysosome inhibition blocks this effect, suggesting a strategy to enhance chemosensitivity.
Baxter JS 2024 [70]	In vitro	BXW7 loss creates a RIF1-dependent synthetic lethality to CDC7 inhibition.	FBXW7-deficient cells show selective sensitivity to CDC7 inhibition; RIF1 silencing rescues this effect, identifying a potential therapeutic vulnerability in FBXW7-mutant cancers.
Hu JL 2019 [71]	In vitro/in vivo (40 cases of CRC tissues and corresponding normal mucosa)	CAF-derived exosomal miR-92a-3p suppresses FBXW7 and MOAP1, activating Wnt/β-catenin signaling to promote EMT, stemness, metastasis, and chemoresistance in CRC.	CAF-derived exosomes enhance CRC metastasis and resistance to 5-FU/L-OHP via miR-92a-3p–mediated suppression of FBXW7 and MOAP1; high serum miR-92a-3p correlates clinically with poor prognosis and chemoresistance in CRC patients.
Li N 2019 [23]	Combined in vitro, ex vivo, and In vivo study (mice)	FBXW7 degrades ZEB2 to suppress EMT, CSC traits, and chemoresistance; its loss stabilizes ZEB2, promoting these processes.	FBXW7 suppresses EMT and CSC formation by degrading ZEB2; its loss enhances chemoresistance and metastasis, while ZEB2 knockdown reverses these effects, highlighting the FBXW7–ZEB2 axis as a key modulator of drug resistance and tumor progression.
Sanchez Burgos L 2022 [72]	In vitro and in vivo (mice)	FBXW7 loss drives multidrug resistance, making cells dependent on the integrated stress response for survival.	FBXW7-deficient, multidrug-resistant cells show hyperactivation of the ISR, making them selectively vulnerable to ISR inhibition both in vitro and in vivo, revealing a therapeutic opportunity to target chemoresistant tumors with defective FBXW7.
Pfohl, U 2022 [73]	In vitro (organoids)	Molecular pattern of CRC and sensitivity to MEK inhibitors	SMAD4 loss and the SFAB signature (SMAD4, FBXW7, ARID1A, BMPR2) predict MEK-inhibitor sensitivity in PDOs, regardless of RAS or BRAF status.

Abbreviations: CRC: colorectal cancer; EMT: epithelial–mesenchymal transition; CSC: cancer stem cells.

**Table 3 ijms-26-11318-t003:** Principals FBXW7 pathways linked to clinical outcomes in CRC.

Molecular Mechanism/Substrate Affected	Biologic Effect	Clinical Correlate/ Observed Finding
EGFR stabilization (impaired ubiquitination)	Increased proliferative signaling; reduced EGF dependency	↓ Sensitivity to anti-EGFR therapy; increased adaptive MAPK/PI3K signaling (organoid and clinical evidence)
Cyclin E1 accumulation	Cell-cycle acceleration; genomic instability	More aggressive biology in some FBXW7-mutant CRCs; worse DFS/OS when co-mutated with TP53
MCL-1 stabilization (impaired degradation)	Anti-apoptotic signaling; survival of stressed clones	Resistance to chemotherapy; MCL-1 inhibition restores drugs sensitivity (preclinical)
NOTCH1/NOTCH signaling dysregulation	Progenitor cell expansion; impaired differentiation	Higher mutation frequency in early-stage, organ-confined CRC; enrichment in AYA CRC and NOTCH-altered tumors
c-MYC stabilization (via SET–FBXW7 interaction	Immunosuppressive microenvironment via CCL5 repression	Immune-excluded MSS CRC phenotype; potential reduced benefit from immunotherapy (preclinical evidence)
HIF-1α stabilization	Upregulation of VEGF-A; angiogenic shift	Possible modulation of responsiveness to anti-angiogenic therapy (clinical data limited)
ZEB2 degradation failure	EMT induction; stemness, migration, invasion	Greater metastatic potential, intrapatient heterogeneity, site-specific differences between primaries/metastases
Interaction with miRNAs (miR-223, miR-27a, miR-182, miR-92b)	Post-transcriptional downregulation of FBXW7	↑ Proliferation, ↑ invasion, ↑ FOXO/Notch/Akt signaling; contribution to chemo- and radio-resistance
circ-FBXW7/exosomal regulation	Modulates FBXW7 expression; miRNA sponging	Exosomal circ-FBXW7reverses oxaliplatin resistance (preclinical)
FBXW7β hotspot mutations (e.g., R465C/H, R479Q)	Variant-specific substrate affinity loss	MSI-H association; better prognosis in specific hotspots; supports variant-level reporting
Loss in colonic stem/progenitor cells	Expansion of undifferentiated cells	Higher prevalence in early-onset CRC; association with right-sided, high-grade tumors
Clonal evolution under treatment (ctDNA emergence)	Resistant clone expansion	Early detection of progression on EGFR therapy; rising FBXW7-mutant ctDNA predicts shorter PFS
High intratumoral heterogeneity	Discordant mutation detection depending on sampling site	Need of re-biopsy of metastatic lesions; stool DNA and ctDNA improve detection

Abbreviations: AYA: adolescent and young adult; CRC: colorectal cancer; DFS: disease free survival; OS: overall survival; PFS: progression free survival; ctDNA: circulating tumor DNA; ↓: decrease; ↑: increase.

## Data Availability

No new data were created or analyzed in this study. Data sharing is not applicable to this article.

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
