# Peer review of "FBXW7 Gene Mutation and Expression in Colorectal Cancer (CRC): A Systematic Review from Molecular Mechanisms to Clinical Translation"

_ijms, 2025, doi:10.3390/ijms262311318_

Round 1
Reviewer 1 Report
Comments and Suggestions for Authors
The systematic review from Arrivi et al. shed light on the role of FBXW7 in colorectal cancer going into detail of the molecular mechanisms with which fbxw7 impacts CRC and its clinical significance. The approach used is interesting and the review very detailed.
Few thighs could be addressed to increase the readability of the text and to make some aspects more focused:
- throughout the text many spaces between words are missing, probably due to the magazine's formatting
- some underlinings remained as typos in the text
- pay attention to the style of the numbered references in the text which is not always consistent
- the title of paragraph 3.2.1 should include CRC to be more clear
- some statements of a more general nature and not strictly related to the focus of the review could be eliminated to lighten some paragraphs. for example the sentences at the lines 227-229, 270-273 and 286-288
- paragraph 3.2.1 could be presented dividing the studies on the basis of the mechanisms identified. For examples i) studies in which FBXW7 has impact on proliferation; ii) on metastatization; etc.
- did you ever search for clinicopathologic association of FBXW7 in cbioportal database? A study including mRNA seq data is available and could be helpful to search also for co-expression data
- the tables in the text are too long. They could easily be streamlined by shortening the texts which are often too detailed for a table that should summarized
Author Response
|
1. Summary |
|
|
|
Thank you very much for taking the time to review this manuscript. Please find the detailed responses below and the corresponding revisions/corrections highlighted/in track changes in the re-submitted files.
|
||
|
2. Point-by-point response to Comments and Suggestions for Authors |
||
|
Comments 1: Throughout the text many spaces between words are missing, probably due to the magazine's formatting. |
||
|
Response 1: Thank you for pointing this out. We agree with this comment. Therefore, we have revised the text to restore the missing spaces and improve overall readability. This adjustment ensures that the content now appears clear, consistent, and faithful to the original intent, regardless of the magazine’s formatting constraints.
|
||
|
Comments 2: Some underlinings remained as typos in the text. |
||
|
Response 2: We agree with this comment. Therefore, we have removed the unintended underlinings that remained in the text, ensuring a clean and consistent formatting throughout the manuscript.
Comments 3: Pay attention to the style of the numbered references in the text which is not always consistent. Response 3: We agree with this comment. Therefore, we have carefully reviewed and corrected the style of all numbered references in the text to ensure full consistency and adherence to the required formatting guidelines.
Comments 4: The title of paragraph 3.2.1 should include CRC to be more clear. Response 4: Thank you for your comment. We have modified it as follows: 'FBXW7 in CRC Tumorigenesis' to make it clearer.
Comments 5: Some statements of a more general nature and not strictly related to the focus of the review could be eliminated to lighten some paragraphs. For example the sentences at the lines 227-229, 270-273 and 286-288. Response 5: We thank the reviewer for this suggestion. We have carefully considered it and removed the more general statements that were not directly related to the focus of the review, specifically the sentences at lines 227–229, 270–273, and 286–288, in order to make the paragraphs more concise and focused.
Comments 6: Paragraph 3.2.1 could be presented dividing the studies on the basis of the mechanisms identified. For examples i) studies in which FBXW7 has impact on proliferation; ii) on metastatization; etc. Response 6: We thank the reviewer for this valuable suggestion. Paragraph 3.2.1 has been completely reorganized, with the studies now divided according to specific mechanisms, such as those in which FBXW7 impacts proliferation, metastasis, and other relevant processes. All changes have been highlighted in red in the revised manuscript.
Comments 7: Did you ever search for clinicopathologic association of FBXW7 in cbioportal database? A study including mRNA seq data is available and could be helpful to search also for co-expression data. Response 7: We thank the reviewer for this insightful comment, which has significantly contributed to improving our manuscript. We have performed the analysis using the cBioPortal database, and the results are reported in Supplementary Table 1. In paragraph 3.2.1, we have added the following statement: 'To further characterize the FBXW7-associated genes in CRC, the co-expression genes predicted by cBioPortal online analysis using Spearman’s correlation are reported in Supplementary Table 1.
Comments 8: The tables in the text are too long. They could easily be streamlined by shortening the texts which are often too detailed for a table that should summarize. Response 8: Thank you for this comment. We have shortened the text in the tables to make them easier to read and consult |
||
Reviewer 2 Report
Comments and Suggestions for Authors
The manuscript is comprehensive and well-referenced but it is dense and lengthy, making it difficult to follow the key points. A more concise structure and critical synthesis of the literature highlighting contradictions and clinical gaps would strengthen clarity and impact. These are my comments:
- The manuscript is comprehensive but overly long, with repeated descriptions of FBXW7-related pathways. Condensing these sections would improve readability.
- No formal risk of bias or study quality assessment is reported. Adding such evaluation would increase the power of the systematic review methodology.
- The integration between molecular mechanisms and clinical findings is weak. A schematic summary linking FBXW7 pathways to clinical outcomes would make the review more cohesive.
- The studies included are highly heterogeneous, ranging from cell lines to small patient cohorts, with inconsistent findings. No standardized methods, IHC cut-offs, or large prospective validations exist. For these reasons, the conclusions are overstated and it would be more accurate to describe FBXW7 as a promising but not yet clinically validated biomarker.
- Some sections are too dense and could benefit from shorter sentences and smoother transitions between molecular and clinical parts.
Author Response
|
1. Summary |
|
|
|
Thank you very much for taking the time to review this manuscript. Please find the detailed responses below and the corresponding revisions/corrections highlighted/in track changes in the re-submitted files.
|
||
|
2. Point-by-point response to Comments and Suggestions for Authors |
||
|
Comments 1: The manuscript is comprehensive but overly long, with repeated descriptions of FBXW7-related pathways. Condensing these sections would improve readability. |
||
|
Response 1: We thank the reviewer for this valuable comment. The entire manuscript has been carefully reviewed and condensed to improve readability, with all changes highlighted in red.
|
||
|
Comments 2: No formal risk of bias or study quality assessment is reported. Adding such evaluation would increase the power of the systematic review methodology. |
||
|
Response 2: We appreciate the reviewer’s constructive comment. A formal risk of bias or study quality assessment was not performed, as the studies included in this review were highly heterogeneous in terms of design, objectives, and endpoints, encompassing both preclinical (in vitro and in vivo) and clinical investigations. The aim of this review was to provide an integrative synthesis of the existing evidence on FBXW7 alterations across the translational continuum rather than to conduct a meta-analysis or compare uniform outcome measures. Nevertheless, methodological rigor was ensured by applying systematic search and selection criteria according to PRISMA guidelines and by independent screening and data extraction by multiple reviewers. We added in Methodology Section reference to bias “A formal risk-of-bias or study quality assessment was not performed due to the heterogeneity of the included studies, which encompassed descriptive preclinical and clinical investigations with differing aims and endpoints.
Comments 3: The integration between molecular mechanisms and clinical findings is weak. A schematic summary linking FBXW7 pathways to clinical outcomes would make the review more cohesive. Response 3: We agree with this comment. We elaborated a brief table according to this comment. In the Discussion chapter we added “Overall, FBXW7 is a promising, though not yet clinically validated, biomarker in CRC. The relationships between its molecular pathways and clinical outcomes are still incomplete and variably supported across studies. To provide a clearer overview, Table 3 summarizes the main FBXW7-related mechanisms and their corresponding clinical implications in CRC” and a table attached below. We also developed a visual abstract that summarize graphically these mechanisms (see Visual Abstract file).
Table 3 |
||
|
Molecular Mechanism / Substrate Affected
|
Biologic Effect
|
Clinical Correlate / Observed Finding
|
|
EGFR stabilization (impaired ubiquitination)
|
Increased proliferative signaling; reduced EGF dependency
|
↓ Sensitivity to anti-EGFR therapy; increased adaptive MAPK/PI3K signaling (organoid and clinical evidence)
|
|
Cyclin E1 accumulation
|
Cell-cycle acceleration; genomic instability
|
More aggressive biology in some FBXW7-mutant CRCs; worse DFS/OS when co-mutated with TP53
|
|
MCL-1 stabilization (impaired degradation)
|
Anti-apoptotic signaling; survival of stressed clones
|
Resistance to chemotherapy; MCL-1 inhibition restores drugs sensitivity (preclinical)
|
|
NOTCH1/NOTCH signaling dysregulation
|
Progenitor cell expansion; impaired differentiation
|
Higher mutation frequency in early-stage, organ-confined CRC; enrichment in AYA CRC and NOTCH-altered tumors
|
|
c-MYC stabilization (via SET–FBXW7 interaction
|
Immunosuppressive microenvironment via CCL5 repression
|
Immune-excluded MSS CRC phenotype; potential reduced benefit from immunotherapy (preclinical evidence)
|
|
HIF-1α stabilization
|
Upregulation of VEGF-A; angiogenic shift
|
Possible modulation of responsiveness to anti-angiogenic therapy (clinical data limited)
|
|
ZEB2 degradation failure |
EMT induction; stemness, migration, invasion |
Greater metastatic potential, intrapatient heterogeneity, site-specific differences between primaries/metastases |
|
Interaction with miRNAs (miR-223, miR-27a, miR-182, miR-92b) |
Post-transcriptional downregulation of FBXW7 |
↑ Proliferation, ↑ invasion, ↑ FOXO/Notch/Akt signaling; contribution to chemo- and radio-resistance |
|
circ-FBXW7/ exosomal regulation |
Modulates FBXW7 expression; miRNA sponging |
Exosomal circ-FBXW7reverses oxaliplatin resistance (preclinical) |
|
FBXW7β hotspot mutations (e.g., R465C/H, R479Q) |
Variant-specific substrate affinity loss |
MSI-H association; better prognosis in specific hotspots; supports variant-level reporting |
|
Loss in in colonic stem/progenitor cells |
Expansion of undifferentiated cells |
Higher prevalence in early-onset CRC; association with right-sided, high-grade tumors |
|
Clonal evolution under treatment (ctDNA emergence) |
Resistant clone expansion |
Early detection of progression on EGFR therapy; rising FBXW7-mutant ctDNA predicts shorter PFS |
|
High intratumoral heterogeneity |
Discordant mutation detection depending on sampling site |
Need of re-biopsy of metastatic lesions; stool DNA and ctDNA improve detection |
Visual Abstract
Comments 4: The studies included are highly heterogeneous, ranging from cell lines to small patient cohorts, with inconsistent findings. No standardized methods, IHC cut-offs, or large prospective validations exist. For these reasons, the conclusions are overstated and it would be more accurate to describe FBXW7 as a promising but not yet clinically validated biomarker.
Response 4: We are very grateful for this advice. We have revised the Discussion chapter and outlined the potential benefits and current limitations of using FBXW7 in CRC clinical practice.
Comments 5: Some sections are too dense and could benefit from shorter sentences and smoother transitions between molecular and clinical parts.
Response 5: We thank the reviewer for this helpful comment. The entire manuscript has been carefully reviewed and condensed to enhance readability, with all changes highlighted in red.
Round 2
Reviewer 2 Report
Comments and Suggestions for Authors
All my previous comments have been fully addressed.